# Demystifying Oversmoothing in Attention-Based Graph Neural Networks

**Xinyi Wu**[1,2]  **Amir Ajorlou**[2]  **Zihui Wu**[3]  **Ali Jadbabaie**[1,2]

[1]Institute for Data, Systems and Society (IDSS), MIT
[2]Laboratory for Information and Decision Systems (LIDS), MIT
[3]Department of Computing and Mathematical Sciences (CMS), Caltech
{xinyiwu,ajorlou,jadbabai}@mit.edu   zwu2@caltech.edu

## Abstract

Oversmoothing in Graph Neural Networks (GNNs) refers to the phenomenon where increasing network depth leads to homogeneous node representations. While previous work has established that Graph Convolutional Networks (GCNs) exponentially lose expressive power, it remains controversial whether the graph attention mechanism can mitigate oversmoothing. In this work, we provide a definitive answer to this question through a rigorous mathematical analysis, by viewing attention-based GNNs as nonlinear time-varying dynamical systems and incorporating tools and techniques from the theory of products of inhomogeneous matrices and the joint spectral radius. We establish that, contrary to popular belief, the graph attention mechanism cannot prevent oversmoothing and loses expressive power exponentially. The proposed framework extends the existing results on oversmoothing for symmetric GCNs to a significantly broader class of GNN models, including random walk GCNs, Graph Attention Networks (GATs) and (graph) transformers. In particular, our analysis accounts for asymmetric, state-dependent and time-varying aggregation operators and a wide range of common nonlinear activation functions, such as ReLU, LeakyReLU, GELU and SiLU.

## 1 Introduction

Graph neural networks (GNNs) have emerged as a powerful framework for learning with graph-structured data [4, 8, 9, 13, 19, 32, 38] and have shown great promise in diverse domains such as molecular biology [45], physics [1] and recommender systems [40]. Most GNN models follow the *message-passing* paradigm [12], where the representation of each node is computed by recursively aggregating and transforming the representations of its neighboring nodes.

One notable drawback of repeated message-passing is *oversmoothing*, which refers to the phenomenon that stacking message-passing GNN layers makes node representations of the same connected component converge to the same vector [5, 18, 19, 24, 26, 31, 41]. As a result, whereas depth has been considered crucial for the success of deep learning in many fields such as computer vision [16], most GNNs used in practice remain relatively shallow and often only have few layers [19, 38, 42]. On the theory side, while previous works have shown that the symmetric Graph Convolution Networks (GCNs) with ReLU and LeakyReLU nonlinearities exponentially lose expressive power, analyzing the oversmoothing phenomenon in other types of GNNs is still an open question [5, 26]. In particular, the question of whether the graph attention mechanism can prevent oversmoothing has not been settled yet. Motivated by the capacity of graph attention to distinguish the importance of different edges in the graph, some works claim that oversmoothing is alleviated in Graph Attention Networks (GATs), heuristically crediting to GATs' ability to learn adaptive node-wise aggregation operators via the attention mechanism [25]. On the other hand, it has been empirically observed that similar to

37th Conference on Neural Information Processing Systems (NeurIPS 2023).

the case of GCNs, oversmoothing seems inevitable for attention-based GNNs such as GATs [31] or (graph) transformers [34]. The latter can be viewed as attention-based GNNs on complete graphs.

In this paper, we provide a definitive answer to this question — attention-based GNNs also lose expressive power exponentially, albeit potentially at a slower exponential rate compared to GCNs. Given that attention-based GNNs can be viewed as nonlinear time-varying dynamical systems, our analysis is built on the theory of products of inhomogeneous matrices [14, 33] and the concept of joint spectral radius [30], as these methods have been long proved effective in the analysis of time-inhomogeneous markov chains and ergodicity of dynamical systems [2, 14, 33]. While classical results only apply to generic one-dimensional linear time-varying systems, we address four major challenges arising in analyzing attention-based GNNs: (1) the aggregation operators computed by attention are state-dependent, in contrast to conventional fixed graph convolutions; (2) the systems are multi-dimensional, which involves the coupling across feature dimensions; (3) the dynamics are nonlinear due to the nonlinear activation function in each layer; (4) the learnable weights and aggregation operators across different layers result in time-varying dynamical systems.

**Below, we highlight our key contributions:**

- As our main contribution, we establish that oversmoothing happens exponentially as model depth increases for attention-based GNNs, resolving the long-standing debate about whether attention-based GNNs can prevent oversmoothing.

- We analyze attention-based GNNs through the lens of nonlinear, time-varying dynamical systems. The strength of our analysis stems from its ability to exploit the inherently common connectivity structure among the typically asymmetric state-dependent aggregation operators at different attentional layers. This enables us to derive rigorous theoretical results on the ergodicity of infinite products of matrices associated with the evolution of node representations across layers. Incorporating results from the theory of products of inhomogeneous matrices and their joint spectral radius, we then establish that oversmoothing happens at an exponential rate for attention-based GNNs from our ergodicity results.

- Our analysis generalizes the existing results on oversmoothing for symmetric GCNs to a significantly broader class of GNN models with asymmetric, state-dependent and time-varying aggregation operators and nonlinear activation functions under general conditions. In particular, our analysis can accommodate a wide range of common nonlinearities such as ReLU, LeakyReLU, and even non-monotone ones like GELU and SiLU. We validate our theoretical results on six real-world datasets with two attention-based GNN architectures and five common nonlinearities.

## 2 Related Work

**Oversmoothing problem in GNNs** Oversmoothing is a well-known problem in deep GNNs, and many techniques have been proposed in order to mitigate it practically [6, 15, 20, 21, 29, 43, 47]. On the theory side, analysis of oversmoothing has only been carried out for the graph convolution case [5, 18, 26, 41]. In particular, by viewing graph convolutions as a form of Laplacian filter, prior works have shown that for GCNs, the node representations within each connected component of a graph will converge to the same value exponentially [5, 26]. However, oversmoothing is also empirically observed in attention-based GNNs such as GATs [31] or transformers [34]. Although some people hypothesize based on heuristics that attention can alleviate oversmoothing [25], a rigorous analysis of oversmoothing in attention-based GNNs remains open [5].

**Theoretical analysis of attention-based GNNs** Existing theoretical results on attention-based GNNs are limited to one-layer graph attention. Recent works in this line include Brody et al. [3] showing that the ranking of attention scores computed by a GAT layer is unconditioned on the query node, and Fountoulakis et al. [11] studying node classification performance of one-layer GATs on a random graph model. More relevantly, Wang et al. [39] made a claim that oversmoothing is asymptotically inevitable in GATs. Aside from excluding nonlinearities in the analysis, there are several flaws in the proof of their main result (Theorem 2). In particular, their analysis assumes the same stationary distribution for all the stochastic matrices output by attention at different layers. This is typically not the case given the state-dependent and time-varying nature of these matrices. In fact, the main challenge in analyzing multi-layer attention lies in the state-dependent and time-varying

nature of these input-output mappings. Our paper offers novel contributions to the research on attention-based GNNs by developing a rich set of tools and techniques for analyzing multi-layer graph attention. This addresses a notable gap in the existing theory, which has primarily focused on one-layer graph attention, and paves the way for future research to study other aspects of multi-layer graph attention.

## 3 Problem Setup

### 3.1 Notations

Let $\mathbb{R}$ be the set of real numbers and $\mathbb{N}$ be the set of natural numbers. We use the shorthands $[n] := \{1, \ldots, n\}$ and $\mathbb{N}_{\geq 0} := \mathbb{N} \cup \{0\}$. We denote the zero-vector of length $N$ by $\mathbf{0} \in \mathbb{R}^N$ and the all-one vector of length $N$ by $\mathbf{1} \in \mathbb{R}^N$. We represent an undirected graph with $N$ nodes by $\mathcal{G} = (A, X)$, where $A \in \{0, 1\}^{N \times N}$ is the *adjacency matrix* and $X \in \mathbb{R}^{N \times d}$ are the *node feature vectors* of dimension $d$. Let $E(\mathcal{G})$ be the set of edges of $\mathcal{G}$. For nodes $i, j \in [N]$, $A_{ij} = 1$ if and only if $i$ and $j$ are connected with an edge in $\mathcal{G}$, i.e., $(i, j) \in E(\mathcal{G})$. For each $i \in [N]$, $X_i \in \mathbb{R}^d$ represents the feature vector for node $i$. We denote the *degree matrix* of $\mathcal{G}$ by $D_{\mathrm{deg}} = \mathrm{diag}(A\mathbf{1})$ and the set of all neighbors of node $i$ by $\mathcal{N}_i$.

Let $\|\cdot\|_2, \|\cdot\|_\infty, \|\cdot\|_F$ be the 2-norm, $\infty$-norm and Frobenius norm, respectively. We use $\|\cdot\|_{\max}$ to denote the matrix max norm, i.e., for a matrix $M \in \mathbb{R}^{m \times n}$, $\|M\|_{\max} := \max_{ij}|M_{ij}|$. We use $\leq_{\mathrm{ew}}$ to denote element-wise inequality. Lastly, for a matrix $M$, we denote its $i^{th}$ row by $M_{i\cdot}$ and $j^{th}$ column by $M_{\cdot j}$.

### 3.2 Graph attention mechanism

We adopt the following definition of graph attention mechanism. Given node representation vectors $X_i$ and $X_j$, we first apply a shared learnable linear transformation $W \in \mathbb{R}^{d \times d'}$ to each node, and then use an attention function $\Psi : \mathbb{R}^{d'} \times \mathbb{R}^{d'} \to \mathbb{R}$ to compute a raw attention coefficient

$$e_{ij} = \Psi(W^\top X_i, W^\top X_j)$$

that indicates the importance of node $j$'s features to node $i$. Then the graph structure is injected into the mechanism by performing masked attention, where for each node $i$, we only compute its attention to its neighbors. To make coefficients easily comparable across different nodes, we normalize $e_{ij}$ among all neighboring nodes $j$ of node $i$ using the softmax function to get the normalized attention coefficients:

$$P_{ij} = \mathrm{softmax}_j(e_{ij}) = \frac{\exp(e_{ij})}{\sum_{k \in \mathcal{N}_i} \exp(e_{ik})}.$$

The matrix $P$, where the entry in the $i^{th}$ row and the $j^{th}$ column is $P_{ij}$, is a row stochastic matrix. We refer to $P$ as an *aggregation operator* in message-passing.

### 3.3 Attention-based GNNs

Having defined the graph attention mechanism, we can now write the update rule of a single graph attentional layer as

$$X' = \sigma(PXW),$$

where $X$ and $X'$ are are the input and output node representations, respectively, $\sigma(\cdot)$ is a pointwise nonlinearity function, and the aggregation operator $P$ is a function of $XW$.

As a result, the output of the $t^{th}$ graph attentional layers can be written as

$$X^{(t+1)} = \sigma(P^{(t)} X^{(t)} W^{(t)}) \qquad t \in \mathbb{N}_{\geq 0}, \tag{1}$$

where $X^{(0)} = X$ is the input node features, $W^{(t)} \in \mathbb{R}^{d' \times d'}$ for $t \in \mathbb{N}$ and $W^{(0)} \in \mathbb{R}^{d \times d'}$. For the rest of this work, without loss of generality, we assume that $d = d'$.

The above definition is based on *single-head* graph attention. *Multi-head* graph attention uses $K \in \mathbb{N}$ weight matrices $W_1, \ldots, W_K$ in each layer and averages their individual single-head outputs [11, 38]. Without loss of generality, we consider single graph attention in our analysis in Section 4, but we note that our results automatically apply to the multi-head graph attention setting since $K$ is finite.

### 3.4 Measure of oversmoothing

We use the following notion of oversmoothing, inspired by the definition proposed in Rusch et al. [31][1]:

**Definition 1.** *For an undirected and connected graph $\mathcal{G}$, $\mu : \mathbb{R}^{N \times d} \to \mathbb{R}_{\geq 0}$ is called a node similarity measure if it satisfies the following axioms:*

1. *$\exists c \in \mathbb{R}^d$ such that $X_i = c$ for all node $i$ if and only if $\mu(X) = 0$, for $X \in \mathbb{R}^{N \times d}$;*

2. *$\mu(X + Y) \leq \mu(X) + \mu(Y)$, for all $X, Y \in \mathbb{R}^{N \times d}$.*

*Then oversmoothing with respect to $\mu$ is defined as the layer-wise convergence of the node-similarity measure $\mu$ to zero, i.e.,*

$$\lim_{t \to \infty} \mu(X^{(t)}) = 0. \tag{2}$$

*We say oversmoothing happens at an exponential rate if there exists constants $C_1$, $C_2 > 0$, such that for any $t \in \mathbb{N}$,*

$$\mu(X^{(t)}) \leq C_1 e^{-C_2 t}. \tag{3}$$

We establish our results on oversmoothing for attention-based GNNs using the following node similarity measure:

$$\mu(X) := \|X - \mathbf{1}\gamma_X\|_F, \text{ where } \gamma_X = \frac{\mathbf{1}^\top X}{N}. \tag{4}$$

**Proposition 1.** $\|X - \mathbf{1}\gamma_X\|_F$ *is a node similarity measure.*

The proof of the above proposition is provided in Appendix B. Other common node similarity measures include the Dirichlet energy [5, 31].[2] Our measure is mathematically equivalent to the measure $\inf_{Y = \mathbf{1}c^\top, c \in \mathbb{R}^d} \{\|X - Y\|_F\}$ defined in Oono and Suzuki [26], but our form is more direct to compute. One way to see the equivalence is to consider the orthogonal projection into the space perpendicular to $\text{span}\{\mathbf{1}\}$, denoted by $B \in \mathbb{R}^{(N-1) \times N}$. Then our definition of $\mu$ satisfies $\|X - \mathbf{1}\gamma_x\|_F = \|BX\|_F$, where the latter quantity is exactly the measure defined in [26].

### 3.5 Assumptions

We make the following assumptions (in fact, quite minimal) in deriving our results:

**A1** The graph $\mathcal{G}$ is connected and non-bipartite.

**A2** The attention function $\Psi(\cdot, \cdot)$ is continuous.

**A3** The sequence $\{\|\prod_{t=0}^{k} |W^{(t)}|\|_{\max}\}_{k=0}^{\infty}$ is bounded.

**A4** The point-wise nonlinear activation function $\sigma(\cdot)$ satisfies $0 \leq \frac{\sigma(x)}{x} \leq 1$ for $x \neq 0$ and $\sigma(0) = 0$.

We note that all of these assumptions are either standard or quite general. Specifically, **A1** is a standard assumption for theoretical analysis on graphs. For graphs with more than one connected components, the same results apply to each connected component. **A1** can also be replaced with requiring the graph $\mathcal{G}$ to be connected and have self-loops at each node. Non-bipartiteness and self-loops both ensure that long products of stochastic matrices corresponding to aggregation operators in different graph attentional layers will eventually become strictly positive.

The assumptions on the GNN architecture **A2** and **A4** can be easily verified for commonly used GNN designs. For example, the attention function $\text{LeakyReLU}(a^\top[W^\top X_i \| W^\top X_j]), a \in \mathbb{R}^{2d'}$ used in the GAT [38], where $[\cdot \| \cdot]$ denotes concatenation, is a specific case that satisfies **A2**. Other architectures that satisfy **A2** include GATv2 [3] and (graph) transformers [37]. As for **A4**, one way to

---

[1]We distinguish the definition of oversmoothing and the rate of oversmoothing. This parallels the notion of stability and its rate in dynamical systems.

[2]In fact, our results are not specific to our choice of node similarity measure $\mu$ and directly apply to any Lipschitz node similarity measure, including the Dirichlet energy under our assumptions. See Remark 2 after Theorem 1.

satisfy it is to have $\sigma$ be 1-Lipschitz and $\sigma(x) \leq 0$ for $x < 0$ and $\sigma(x) \geq 0$ for $x > 0$. Then it is easy to verify that most of the commonly used nonlinear activation functions such as ReLU, LeakyReLU, GELU, SiLU, ELU, tanh all satisfy **A4**.

Lastly, **A3** is to ensure boundedness of the node representations' trajectories $X^{(t)}$ for all $t \in \mathbb{N}_{\geq 0}$. Such regularity assumptions are quite common in the asymptotic analysis of dynamical systems, as is also the case for the prior works analyzing oversmoothing in symmetric GCNs [5, 26].

## 4 Main Results

In this section, we lay out a road-map for deriving our main results, highlighting the key ideas of the proofs. The complete proofs are provided in the Appendices.

We start by discussing the dynamical system formulation of attention-based GNNs in Section 4.1. By showing the boundedness of the node representations' trajectories, we prove the existence of a common connectivity structure among aggregation operators across different graph attentional layers in Section 4.2. This implies that graph attention cannot fundamentally change the graph connectivity, a crucial property that will eventually lead to oversmoothing. In Section 4.3, we develop a framework for investigating the asymptotic behavior of attention-based GNNs by introducing the notion of ergodicity and its connections to oversmoothing. Then utilizing our result on common connectivity structure among aggregation operators, we establish ergodicity results for the systems associated with attention-based GNNs. In Section 4.4, we introduce the concept of the joint spectral radius for a set of matrices [30] and employ it to deduce exponential convergence of node representations to a common vector from our ergodicity results. Finally, we present our main result on oversmoothing in attention-based GNNs in Section 4.5 and comment on oversmoothing in GCNs in comparison with attention-based GNNs in Section 4.6.

### 4.1 Attention-based GNNs as nonlinear time-varying dynamical systems

The theory of dynamical systems concerns the evolution of some state of interest over time. By viewing the model depth $t$ as the time variable, the input-output mapping at each graph attentional layer $X^{(t+1)} = \sigma(P^{(t)} X^{(t)} W^{(t)})$ describes a nonlinear time-varying dynamical system. The attention-based aggregation operator $P^{(t)}$ is state-dependent as it is a function of $X^{(t)} W^{(t)}$. Given the notion of oversmoothing defined in Section 3.4, we are interested in characterizing behavior of $X^{(t)}$ as $t \to \infty$.

If the activation function $\sigma(\cdot)$ is the identity map, then repeated application of (1) gives

$$X^{(t+1)} = P^{(t)} \dots P^{(0)} X W^{(0)} \dots W^{(t)}.$$

The above linear form would enable us to leverage the rich literature on the asymptotic behavior of the products of inhomogeneous row-stochastic matrices (see, e.g., [14, 33]) in analyzing the long-term behavior of attention-based GNNs. Such a neat expansion, however, is not possible when dealing with a nonlinear activation function $\sigma(\cdot)$. To find a remedy, let us start by observing that element-wise application of $\sigma$ to a vector $y \in \mathbb{R}^d$ can be written as

$$\sigma(y) = \operatorname{diag}\left(\frac{\sigma(y)}{y}\right) y, \tag{5}$$

where $\operatorname{diag}\left(\frac{\sigma(y)}{y}\right)$ is a diagonal matrix with $\sigma(y_i)/y_i$ on the $i^{th}$ diagonal entry. Defining $\sigma(0)/0 := \sigma'(0)$ or 1 if the derivative does not exist along with the assumption $\sigma(0) = 0$ in **A4**, it is easy to check that the above identity still holds for vectors with zero entries.

We can use (5) to write the $i^{th}$ column of $X^{(t+1)}$ as

$$X_{\cdot i}^{(t+1)} = \sigma(P^{(t)}(X^{(t)} W^{(t)})_{\cdot i}) = D_i^{(t)} P^{(t)} (X^{(t)} W^{(t)})_{\cdot i} = D_i^{(t)} P^{(t)} \sum_{j=1}^{d} W_{ji}^{(t)} X_{\cdot j}^{(t)}, \tag{6}$$

where $D_i^{(t)}$ is a diagonal matrix. It follows from the assumption on the nonlinearities **A4** that

$$\operatorname{diag}(\mathbf{0}) \leq_{\text{ew}} D_i^{(t)} \leq_{\text{ew}} \operatorname{diag}(\mathbf{1}).$$

We define $\mathcal{D}$ to be the set of all possible diagonal matrices $D_i^{(t)}$ satisfying the above inequality:

$$\mathcal{D} := \{\mathrm{diag}(\mathbf{d}) : \mathbf{d} \in \mathbb{R}^N, \mathbf{0} \leq_{\mathrm{ew}} \mathbf{d} \leq_{\mathrm{ew}} \mathbf{1}\}.$$

Using (6) recursively, we arrive at the following formulation for $X_{\cdot i}^{(t+1)}$:

$$X_{\cdot i}^{(t+1)} = \sum_{j_{t+1}=i,\, (j_t,\ldots,j_0)\in[d]^{t+1}} \left(\prod_{k=0}^{t} W_{j_k j_{k+1}}^{(k)}\right) D_{j_{t+1}}^{(t)} P^{(t)} \ldots D_{j_1}^{(0)} P^{(0)} X_{\cdot j_0}^{(0)}. \qquad (7)$$

## 4.2 Common connectivity structure among aggregation operators across different layers

We can use the formulation in (7) to show the boundedness of the node representations' trajectories $X^{(t)}$ for all $t \in \mathbb{N}_{\geq 0}$, which in turn implies the boundedness of the input to graph attention in each layer, $X^{(t)}W^{(t)}$.

**Lemma 1.** *Under assumptions* **A3**-**A4***, there exists $C > 0$ such that $\|X^{(t)}\|_{\max} \leq C$ for all $t \in \mathbb{N}_{\geq 0}$.*

For a continuous $\Psi(\cdot, \cdot)$[3], the following lemma is a direct consequence of Lemma 1, suggesting that the graph attention mechanism cannot fundamentally change the connectivity pattern of the graph.

**Lemma 2.** *Under assumptions* **A2**-**A4***, there exists $\epsilon > 0$ such that for all $t \in \mathbb{N}_{\geq 0}$ and for any $(i,j) \in E(\mathcal{G})$, we have $P_{ij}^{(t)} \geq \epsilon$.*

One might argue that Lemma 2 is an artifact of the continuity of the softmax function. The softmax function is, however, often favored in attention mechanisms because of its trainability in back propagation compared to discontinuous alternatives such as hard thresholding. Besides trainability issues, it is unclear on a conceptual level whether it is reasonable to absolutely drop an edge from the graph as is the case for hard thresholding. Lemma 2 is an important step towards the main convergence result of this work, which states that all the nodes will converge to the same representation vector at an exponential rate. We define the family of row-stochastic matrices satisfying Lemma 2 below.

**Definition 2.** *Let $\epsilon > 0$. We define $\mathcal{P}_{\mathcal{G},\epsilon}$ to be the set of row-stochastic matrices satisfying the following conditions:*

1. *$\epsilon \leq P_{ij} \leq 1$, if $(i,j) \in E(\mathcal{G})$,*

2. *$P_{ij} = 0$, if $(i,j) \notin E(\mathcal{G})$,*

## 4.3 Ergodicity of infinite products of matrices

*Ergodicity*, in its most general form, deals with the long-term behavior of dynamical systems. The oversmoothing phenomenon in GNNs defined in the sense of (2) concerns the convergence of all rows of $X^{(t)}$ to a common vector. To this end, we define ergodicity in our analysis as the convergence of infinite matrix products to a rank-one matrix with identical rows.

**Definition 3** (Ergodicity). *Let $B \in \mathbb{R}^{(N-1)\times N}$ be the orthogonal projection onto the space orthogonal to $\mathrm{span}\{\mathbf{1}\}$. A sequence of matrices $\{M^{(n)}\}_{n=0}^{\infty}$ is ergodic if*

$$\lim_{t\to\infty} B \prod_{n=0}^{t} M^{(n)} = 0.$$

We will take advantage of the following properties of the projection matrix $B$ already established in Blondel et al. [2]:

1. $B\mathbf{1} = 0$;
2. $\|Bx\|_2 = \|x\|_2$ for $x \in \mathbb{R}^N$ if $x^\top \mathbf{1} = 0$;
3. Given any row-stochastic matrix $P \in \mathbb{R}^{N\times N}$, there exists a unique matrix $\tilde{P} \in \mathbb{R}^{(N-1)\times(N-1)}$ such that $BP = \tilde{P}B$.

---

[3]More generally, for $\Psi(\cdot, \cdot)$ that outputs bounded attention scores for bounded inputs.

We can use the existing results on the ergodicity of infinite products of inhomogeneous stochastic matrices [14, 33] to show that any sequence of matrices in $\mathcal{P}_{\mathcal{G},\epsilon}$ is ergodic.

**Lemma 3.** *Fix $\epsilon > 0$. Consider a sequence of matrices $\{P^{(t)}\}_{t=0}^{\infty}$ in $\mathcal{P}_{\mathcal{G},\epsilon}$. That is, $P^{(t)} \in \mathcal{P}_{\mathcal{G},\epsilon}$ for all $t \in \mathbb{N}_{\geq 0}$. Then $\{P^{(t)}\}_{t=0}^{\infty}$ is ergodic.*

The main proof strategy for Lemma 3 is to make use of the Hilbert projective metric and the Birkhoff contraction coefficient. These are standard mathematical tools to prove that an infinite product of inhomogeneous stochastic matrices is ergodic. We refer interested readers to the textbooks [14, 33] for a comprehensive study of these subjects.

Despite the nonlinearity of $\sigma(\cdot)$, the formulation (7) enables us to express the evolution of the feature vector trajectories as a weighted sum of products of matrices of the form $DP$ where $D \in \mathcal{D}$ and $P \in \mathcal{P}_{\mathcal{G},\epsilon}$. We define the set of such matrices as

$$\mathcal{M}_{\mathcal{G},\epsilon} := \{DP : D \in \mathcal{D}, P \in \mathcal{P}_{\mathcal{G},\epsilon}\}.$$

A key step in proving oversmoothing for attention-based GNNs under our assumptions is to show the ergodicity of the infinite products of matrices in $\mathcal{M}_{\mathcal{G},\epsilon}$. In what follows, we lay out the main ideas of the proof, and refer readers to Appendix F for the details.

Consider a sequence $\{D^{(t)}P^{(t)}\}_{t=0}^{\infty}$ in $\mathcal{M}_{\mathcal{G},\epsilon}$, that is, $D^{(t)}P^{(t)} \in \mathcal{M}_{\mathcal{G},\epsilon}$ for all $t \in \mathbb{N}_{\geq 0}$. For $t_0 \leq t_1$, define

$$Q_{t_0,t_1} := D^{(t_1)}P^{(t_1)}\ldots D^{(t_0)}P^{(t_0)}, \qquad \delta_t = \|D^{(t)} - I_N\|_{\infty},$$

where $I_N$ denotes the $N \times N$ identity matrix. The common connectivity structure among $P^{(t)}$'s established in Section 4.2 allows us to show that long products of matrices $DP$ from $\mathcal{M}_{\mathcal{G},\epsilon}$ will eventually become a contraction in $\infty$-norm. More precisely, we can show that there exists $T \in \mathbb{N}$ and $0 < c < 1$ such that for all $t \in \mathbb{N}_{\geq 0}$,

$$\|Q_{t,t+T}\|_{\infty} \leq 1 - c\delta_t.$$

Next, define $\beta_k := \prod_{t=0}^{k}(1 - c\delta_t)$ and let $\beta := \lim_{k\to\infty} \beta_k$. Note that $\beta$ is well-defined because the partial product is non-increasing and bounded from below. We can use the above contraction property to show the following key lemma.

**Lemma 4.** *Let $\beta_k := \prod_{t=0}^{k}(1 - c\delta_t)$ and $\beta := \lim_{k\to\infty} \beta_k$.*

1. *If $\beta = 0$, then $\lim_{k\to\infty} Q_{0,k} = 0$;*

2. *If $\beta > 0$, then $\lim_{k\to\infty} BQ_{0,k} = 0$.*

The ergodicity of sequences of matrices in $\mathcal{M}_{\mathcal{G},\epsilon}$ immediately follows from Lemma 4, which in turn implies oversmoothing as defined in (2).

**Lemma 5.** *Any sequence $\{D^{(t)}P^{(t)}\}_{t=0}^{\infty}$ in $\mathcal{M}_{\mathcal{G},\epsilon}$ is ergodic.*

**Remark** The proof techniques developed in [5, 26] are restricted to symmetric matrices hence cannot be extended to more general family of GNNs, as they primarily rely on matrix norms for convergence analysis. Analyses solely using matrix norms are often too coarse to get meaningful results when it comes to asymmetric matrices. For instance, while the matrix 2-norm and matrix eigenvalues are directly related for symmetric matrices, the same does not generally hold for asymmetric matrices. Our analysis, on the other hand, exploits the inherently common connectivity structure among these matrices in deriving the ergodicity results in Lemma 3-5.

## 4.4 Joint spectral radius

Using the ergodicity results in the previous section, we can establish that oversmoothing happens in attention-based GNNs. To show that oversmoothing happens at an exponential rate, we introduce the notion of *joint spectral radius*, which is a generalization of the classical notion of spectral radius of a single matrix to a set of matrices [7, 30]. We refer interested readers to the textbook [17] for a comprehensive study of the subject.

**Definition 4** (Joint Spectral Radius). *For a collection of matrices $\mathcal{A}$, the joint spectral radius $\mathrm{JSR}(\mathcal{A})$ is defined to be*

$$\mathrm{JSR}(\mathcal{A}) = \limsup_{k \to \infty} \sup_{A_1, A_2, \ldots, A_k \in \mathcal{M}} \|A_1 A_2 \ldots A_k\|^{\frac{1}{k}},$$

*and it is independent of the norm used.*

In plain words, the joint spectral radius measures the maximal asymptotic growth rate that can be obtained by forming long products of matrices taken from the set $\mathcal{A}$. To analyze the convergence rate of products of matrices in $\mathcal{M}_{\mathcal{G},\epsilon}$ to a rank-one matrix with identical rows, we treat the two cases of linear and nonlinear activation functions, separately.

For the linear case, where $\sigma(\cdot)$ is the identity map, we investigate the dynamics induced by $P^{(t)}$'s on the subspace orthogonal to $\mathrm{span}\{\mathbf{1}\}$ and use the third property of the orthogonal projection $B$ established in Section 4.3 to write $BP_1 P_2 \ldots P_k = \tilde{P}_1 \tilde{P}_2 \ldots \tilde{P}_k B$, where each $\tilde{P}_i$ is the unique matrix in $\mathbb{R}^{(N-1) \times (N-1)}$ that satisfies $BP_i = \tilde{P}_i B$. Let us define

$$\tilde{\mathcal{P}}_{\mathcal{G},\epsilon} := \{\tilde{P} : BP = \tilde{P}B, P \in \mathcal{P}_{\mathcal{G},\epsilon}\}.$$

We can use Lemma 3 to show that the joint spectral radius of $\tilde{\mathcal{P}}_{\mathcal{G},\epsilon}$ is strictly less than 1.

**Lemma 6.** *Let $0 < \epsilon < 1$. Under assumptions **A1**-**A4**, $\mathrm{JSR}(\tilde{\mathcal{P}}_{\mathcal{G},\epsilon}) < 1$.*

For the nonlinear case, let $0 < \delta < 1$ and define

$$\mathcal{D}_\delta := \{\mathrm{diag}(\mathbf{d}) : \mathbf{d} \in \mathbb{R}^N, \mathbf{0} \leq_{\mathrm{ew}} \mathbf{d} \leq_{\mathrm{ew}} \delta\}, \qquad \mathcal{M}_{\mathcal{G},\epsilon,\delta} := \{\mathrm{DP} : \mathrm{D} \in \mathcal{D}_\delta, \mathrm{P} \in \mathcal{P}_{\mathcal{G},\epsilon}\}.$$

Then again, using the ergodicity result from the previous section, we establish that the joint spectral radius of $\mathcal{M}_{\mathcal{G},\epsilon,\delta}$ is also less than 1.

**Lemma 7.** *Let $0 < \epsilon, \delta < 1$. Under assumptions **A1**-**A4**, $\mathrm{JSR}(\mathcal{M}_{\mathcal{G},\epsilon,\delta}) < 1$.*

The above lemma is specifically useful in establishing exponential rate for oversmoothing when dealing with nonlinearities for which Assumption 4 holds in the strict sense, i.e. $0 \leq \frac{\sigma(x)}{x} < 1$ (e.g., GELU and SiLU nonlinearities). Exponential convergence, however, can still be established under a weaker requirement, making it applicable to ReLU and Leaky ReLU, as we will see in Theorem 1.

It follows from the definition of the joint spectral radius that if $\mathrm{JSR}(\mathcal{A}) < 1$, for any $\mathrm{JSR}(\mathcal{A}) < q < 1$, there exists a $C$ for which

$$\|A_1 A_2 \ldots A_k y\| \leq Cq^k \|y\| \tag{8}$$

for all $y \in \mathbb{R}^{N-1}$ and $A_1, A_2, \ldots, A_k \in \mathcal{A}$.

## 4.5 Main Theorem

Applying (8) to the recursive expansion of $X_{\cdot i}^{(t+1)}$ in (7) using the 2-norm, we can prove the exponential convergence of $\mu(X^{(t)})$ to zero for the similarity measure $\mu(\cdot)$ defined in (4), which in turn implies the convergence of node representations to a common representation at an exponential rate. This completes the proof of the main result of this paper, which states that oversmoothing defined in (2) is unavoidable for attention-based GNNs, and that an exponential convergence rate can be attained under general conditions.

**Theorem 1.** *Under assumptions **A1**-**A4**,*

$$\lim_{t \to \infty} \mu(X^{(t)}) = 0,$$

*indicating oversmoothing happens asymptotically in attention-based GNNs with general nonlinearities. In addition, if*

- *(linear) $\sigma(\cdot)$ is the identity map, or*

- *(nonlinear) there exists $K \in \mathbb{N}$ and $0 < \delta < 1$ for which the following holds: For all $m \in \mathbb{N}_{\geq 0}$, there is $n_m \in \{0\} \cup [K-1]$ such that for any $c \in [d]$, $\sigma(X_{r_c c}^{(mK+n_m)})/X_{r_c c}^{(mK+n_m)} \leq \delta$ for some $r_c \in [N]$,* $(\star)$

*then there exists $q < 1$ and $C_1(q) > 0$ such that*

$$\mu(X^{(t)}) \leq C_1 q^t, \forall t \geq 0.$$

As a result, node representations $X^{(t)}$ exponentially converge to the same value as the model depth $t \to \infty$.

Theorem 1 establishes that oversmoothing is asymptotically inevitable for attention-based GNNs with general nonlinearities. Despite similarity-based importance assigned to different nodes via the aggregation operator $P^{(t)}$, *such attention-based mechanisms are yet unable to fundamentally change the connectivity structure of $P^{(t)}$*, resulting in node representations converging to a common vector. Our results hence indirectly support the emergence of alternative ideas for changing the graph connectivity structure such as edge-dropping [15, 29] or graph-rewiring [21], in an effort to mitigate oversmoothing.

**Remark 1.** *For nonlinearities such as SiLU or GELU, the condition $(\star)$ is automatically satisfied under A3-A4. For ReLU and LeakyReLU, this is equivalent to requiring that there exists $K \in \mathbb{N}$ such that for all $m \in \mathbb{N}_{\geq 0}$, there exists $n_m \in \{0\} \cup [K-1]$ where for any $c \in [d]$, $X_{r_c c}^{(mK+n_m)} < 0$ for some $r_c \in [d]$.*

**Remark 2.** *We note that our results are not specific to the choice of node similarity measure $\mu(X) = \|X - \mathbf{1}_{\gamma_X}\|_F$ considered in our analysis. In fact, exponential convergence of any other Lipschitz node similarity measure $\mu'$ to $0$ is a direct corollary of Theorem 1. To see this, observe that for a node similarity measure $\mu'$ with a Lipschitz constant $L$, it holds that*

$$\mu'(X) = |\mu'(X) - \mu'(\mathbf{1}_{\gamma_X})| \leq L\|X - \mathbf{1}_{\gamma_X}\|_F = L\mu(X).$$

*In particular, the Dirichlet energy is Lipschitz given that the input $X$ has a compact domain, established in Lemma 1. Hence our theory directly implies the exponential convergence of Dirichlet energy.*

### 4.6 Comparison with the GCN

Computing or approximating the joint spectral radius for a given set of matrices is known to be hard in general [36], yet it is straightforward to lower bound $\mathrm{JSR}(\tilde{\mathcal{P}}_{\mathcal{G},\epsilon})$ as stated in the next proposition.

**Proposition 2.** *Let $\lambda$ be the second largest eigenvalue of $D_{\deg}^{-1/2} A D_{\deg}^{-1/2}$. Then under assumptions A1-A4, it holds that $\lambda \leq \mathrm{JSR}(\tilde{\mathcal{P}}_{\mathcal{G},\epsilon})$.*

In the linear case, the upper bound $q$ on the convergence rate that we get for graph attention in Theorem 1 is lower bounded by $\mathrm{JSR}(\tilde{\mathcal{P}}_{\mathcal{G},\epsilon})$. A direct consequence of the above result is that $q$ is at least as large as $\lambda$. On the other hand, previous work has already established that in the graph convolution case, the convergence rate of $\mu(X^{(t)})$ is $O(\lambda^t)$ [5, 26]. It is thus natural to expect attention-based GNNs to potentially have better expressive power at finite depth than GCNs, even though they both inevitably suffer from oversmoothing. This is also evident from the numerical experiments that we present in the next section.

## 5 Numerical Experiments

In this section, we validate our theoretical findings via numerical experiments using the three commonly used homophilic benchmark datasets: Cora, CiteSeer, and PubMed [44] and the three commonly used heterophilic benchmark datasets: Cornell, Texas, and Wisconsin [28]. We note that our theoretical results are developed for generic graphs and thus hold for datasets exhibiting either homophily or heterophily and even those that are not necessarily either of the two. More details about the experiments are provided in Appendix K.

For each dataset, we trained a 128-layer single-head GAT and a 128-layer GCN with the random walk graph convolution $D_{\deg}^{-1} A$, each having 32 hidden dimensions and trained using the standard features and splits. The GCN with the random walk graph convolution is a special type of attention-based GNNs where the attention function is constant. For each GNN model, we considered various nonlinear activation functions: ReLU, LeakyReLU (with three different negative slope values: 0.01, 0.4 and 0.8) and GELU. Here, we chose GELU as an illustration of the generality of our assumption on nonlinearities, covering even non-monotone activation functions such as GELU. We ran each experiment 10 times. Figure 1 shows the evolution of $\mu(X^{(t)})$ in log-log scale on the largest connected component of each graph as we forward pass the input $X$ into a trained model. The solid curve is the average over 10 runs and the band indicates one standard deviation around the average.

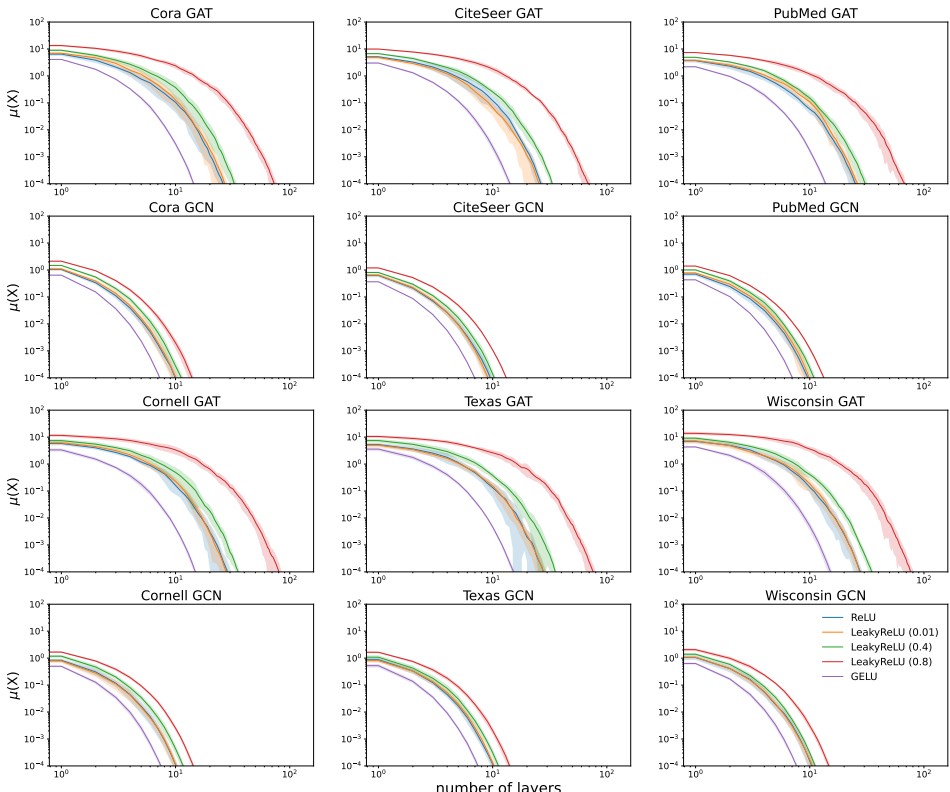

Figure 1: Evolution of $\mu(X^{(t)})$ (in log-log scale) on the largest connected component of each dataset (top 2 rows: homophilic graphs; bottom 2 rows: heterophilic graphs). Oversmoothing happens exponentially in both GCNs and GATs with the rates varying depending on the choice of activation function. Notably, GCNs demonstrate faster rates of oversmoothing compared to GATs.

We observe that, as predicted by our theory, oversmoothing happens at an exponential rate for both GATs and GCNs, regardless of the choice of nonlinear activation functions in the GNN architectures. Notably, GCNs exhibit a significantly faster rate of oversmoothing compared to GATs. This aligns the observation made in Section 4.6, expecting a potentially better expressive power for GATs than GCNs at finite depth. Furthermore, the exponential convergence rate of oversmoothing varies among GNNs with different nonlinear activation functions. From a theory perspective, as different activation functions constitute different subsets of $\mathcal{M}_{\mathcal{G},\epsilon}$ and different sets of matrices have different joint spectral radii, it is not surprising that the choice of nonlinear activation function would affect the convergence rate. In particular, among the nonlinearties we considered, ReLU in fact magnifies oversmoothing the second most. As a result, although ReLU is often the default choice for the standard implementation of many GNN architectures [10, 19], one might wish to consider switching to other nonliearities to better mitigate oversmoothing.

## 6 Conclusion

Oversmoothing is one of the central challenges in developing more powerful GNNs. In this work, we reveal new insights on oversmoothing in attention-based GNNs by rigorously providing a negative answer to the open question of whether graph attention can implicitly prevent oversmoothing. By analyzing the graph attention mechanism within the context of nonlinear time-varying dynamical systems, we establish that attention-based GNNs lose expressive power exponentially as model depth increases.

We upper bound the convergence rate for oversmoothing under very general assumptions on the nonlinear activation functions. One may try to tighten the bounds by refining the analysis separately for each of the commonly used activation functions. Future research should also aim to improve the design of graph attention mechanisms based on our theoretical insights and utilize our analysis techniques to study other aspects of multi-layer graph attention.

## Acknowledgments

Xinyi Wu would like to thank Jennifer Tang and William Wang for helpful discussions. The authors are grateful to Zhijian Zhuo and Yifei Wang for identifying an error in an earlier draft of the paper, thank the anonymous reviewers for providing valuable feedback, and acknowledge the MIT SuperCloud and Lincoln Laboratory Supercomputing Center for providing computing resources that have contributed to the research results reported within this paper.

This research has been supported in part by ARO MURI W911NF-19-0217, ONR N00014-20-1-2394, and the MIT-IBM Watson AI Lab.

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

# A Basic Facts about Matrix Norms

In this section, we list some basic facts about matrix norms that will be helpful in comprehending the subsequent proofs.

## A.1 Matrix norms induced by vector norms

Suppose a vector norm $\|\cdot\|_\alpha$ on $\mathbb{R}^n$ and a vector norm $\|\cdot\|_\beta$ on $\mathbb{R}^m$ are given. Any matrix $M \in \mathbb{R}^{m \times n}$ induces a linear operator from $\mathbb{R}^n$ to $\mathbb{R}^m$ with respect to the standard basis, and one defines the corresponding *induced norm* or *operator norm* by

$$\|M\|_{\alpha,\beta} = \sup \left\{ \frac{\|Mv\|_\beta}{\|v\|_\alpha}, v \in \mathbb{R}^n, v \neq \mathbf{0} \right\}.$$

If the $p$-norm for vectors ($1 \leq p \leq \infty$) is used for both spaces $\mathbb{R}^n$ and $\mathbb{R}^m$, then the corresponding operator norm is

$$\|M\|_p = \sup_{v \neq \mathbf{0}} \frac{\|Mv\|_p}{\|v\|_p}.$$

The matrix 1-norm and $\infty$-norm can be computed by

$$\|M\|_1 = \max_{1 \leq j} \sum_{i=1}^{m} |M_{ij}|,$$

that is, the maximum absolute column sum of the matrix $M$;

$$\|M\|_\infty = \max_{1 \leq m} \sum_{j=1}^{n} |M_{ij}|,$$

that is, the maximum absolute row sum of the matrix $M$.

**Remark** In the special case of $p = 2$, the induced matrix norm $\|\cdot\|_2$ is called the *spectral norm*, and is equal to the largest singular value of the matrix.

For square matrices, we note that the name "spectral norm" does not imply the quantity is directly related to the spectrum of a matrix, unless the matrix is symmetric.

**Example** We give the following example of a stochastic matrix $P$, whose spectral radius is 1, but its spectral norm is greater than 1.

$$P = \begin{bmatrix} 0.9 & 0.1 \\ 0.25 & 0.75 \end{bmatrix} \qquad \|P\|_2 \approx 1.0188$$

## A.2 Matrix $(p, q)$-norms

The Frobenius norm of a matrix $M \in \mathbb{R}^{m \times n}$ is defined as

$$\|M\|_F = \sqrt{\sum_{j=1}^{n} \sum_{i=1}^{m} |M_{ij}|^2},$$

and it belongs to a family of entry-wise matrix norms: for $1 \leq p, q \leq \infty$, the matrix $(p, q)$-norm is defined as

$$\|M\|_{p,q} = \left( \sum_{j=1}^{n} \left( \sum_{i=1}^{m} |M_{ij}|^p \right)^{q/p} \right)^{1/q}.$$

The special case $p = q = 2$ is the Frobenius norm $\|\cdot\|_F$, and $p = q = \infty$ yields the max norm $\|\cdot\|_{\max}$.

## A.3 Equivalence of norms

For any two matrix norms $\|\cdot\|_\alpha$ and $\|\cdot\|_\beta$, we have that for all matrices $M \in \mathbb{R}^{m \times n}$,

$$r\|M\|_\alpha \le \|M\|_\beta \le s\|M\|_\alpha$$

for some positive numbers $r$ and $s$. In particular, the following inequality holds for the 2-norm $\|\cdot\|_2$ and the $\infty$-norm $\|\cdot\|_\infty$:

$$\frac{1}{\sqrt{n}}\|M\|_\infty \le \|M\|_2 \le \sqrt{m}\|M\|_\infty.$$

# B Proof of Proposition 1

It is straightforward to check that $\|X - \mathbf{1}\gamma_X\|_F$ satisfies the two axioms of a node similarity measure:

1. $\|X - \mathbf{1}\gamma_X\|_F = 0 \iff X = \mathbf{1}\gamma_X \iff X_i = \gamma_X$ for all node $i$.

2. Let $\gamma_X = \frac{\mathbf{1}^\top X}{N}$ and $\gamma_Y = \frac{\mathbf{1}^\top Y}{N}$, then $\gamma_X + \gamma_Y = \frac{\mathbf{1}^\top (X+Y)}{N} = \gamma_{X+Y}$. So

$$\begin{aligned}
\mu(X+Y) &= \|(X+Y) - \mathbf{1}(\gamma_X + \gamma_Y)\|_F = \|X - \mathbf{1}\gamma_X + Y - \mathbf{1}\gamma_Y\|_F \\
&\le \|X - \mathbf{1}\gamma_X\|_F + \|Y - \mathbf{1}\gamma_Y\|_F \\
&= \mu(X) + \mu(Y).
\end{aligned}$$

# C Proof of Lemma 1

According to the formulation (7):

$$X_{\cdot i}^{(t+1)} = \sum_{j_{t+1}=i,\,(j_t,\dots,j_0)\in[d]^{t+1}} \left(\prod_{k=0}^{t} W_{j_k j_{k+1}}^{(k)}\right) D_{j_{t+1}}^{(t)} P^{(t)}\dots D_{j_1}^{(0)} P^{(0)} X_{\cdot j_0}^{(0)},$$

we thus obtain that

$$\begin{aligned}
\|X_{\cdot i}^{(t+1)}\|_\infty &= \left\| \sum_{j_{t+1}=i,\,(j_t,\dots,j_0)\in[d]^{t+1}} \left(\prod_{k=0}^{t} W_{j_k j_{k+1}}^{(k)}\right) D_{j_{t+1}}^{(t)} P^{(t)}\dots D_{j_1}^{(0)} P^{(0)} X_{\cdot j_0}^{(0)} \right\|_\infty \\
&\le \sum_{j_{t+1}=i,\,(j_t,\dots,j_0)\in[d]^{t+1}} \left(\prod_{k=0}^{t} \left|W_{j_k j_{k+1}}^{(k)}\right|\right) \left\|D_{j_{t+1}}^{(t)} P^{(t)}\dots D_{j_1}^{(0)} P^{(0)}\right\|_\infty \left\|X_{\cdot j_0}^{(0)}\right\|_\infty \\
&\le \sum_{j_{t+1}=i,\,(j_t,\dots,j_0)\in[d]^{t+1}} \left(\prod_{k=0}^{t} \left|W_{j_k j_{k+1}}^{(k)}\right|\right) \left\|X_{\cdot j_0}^{(0)}\right\|_\infty \\
&\le C_0 \left(\sum_{j_{t+1}=i,\,(j_t,\dots,j_0)\in[d]^{t+1}} \left(\prod_{k=0}^{t} \left|W_{j_k j_{k+1}}^{(k)}\right|\right)\right) \\
&= C_0 \|(|W^{(0)}|\dots|W^{(t)}|)_{\cdot i}\|_1,
\end{aligned}$$

where $C_0$ equals the maximal entry in $|X^{(0)}|$.

The assumption **A3** implies that there exists $C' > 0$ such that for all $t \in \mathbb{N}_{\ge 0}$ and $i \in [d]$,

$$\|(|W^{(0)}|\dots|W^{(t)}|)_{\cdot i}\|_1 \le C'N.$$

Hence there exists $C'' > 0$ such that for all $t \in \mathbb{N}_{\ge 0}$ and $i \in [d]$, we have

$$\|X_{\cdot i}^{(t)}\|_\infty \le C'',$$

proving the existence of $C > 0$ such that $\|X^{(t)}\|_{\max} \le C$ for all $t \in \mathbb{N}_{\ge 0}$.

# D Proof of Lemma 2

Lemma 2 is a direct corollary of Lemma 1 and the assumption that $\Psi(\cdot, \cdot)$ assigns bounded attention scores to bounded inputs.

# E Proof of Lemma 3

## E.1 Auxiliary results

We make use of the following sufficient condition for the ergodicity of the infinite products of row-stochastic matrices.

**Lemma 8** (Corollary 5.1 [14]). *Consider a sequence of row-stochastic matrices $\{S^{(t)}\}_{t=0}^{\infty}$. Let $a_t$ and $b_t$ be the smallest and largest entries in $S^{(t)}$, respectively. If $\sum_{t=0}^{\infty} \frac{a_t}{b_t} = \infty$, then $\{S^{(t)}\}_{t=0}^{\infty}$ is ergodic.*

In order to make use of the above result, we first show that long products of $P^{(t)}$'s from $\mathcal{P}_{\mathcal{G},\epsilon}$ will eventually become strictly positive. For $t_0 \leq t_1$, we denote

$$P^{(t_1:t_0)} = P^{(t_1)} \cdots P^{(t_0)}.$$

**Lemma 9.** *Under the assumption **A1**, there exist $T \in \mathbb{N}$ and $c > 0$ such that for all $t_0 \geq 0$,*

$$c \leq P_{ij}^{(t_0+T:t_0)} \leq 1, \forall 1 \leq i,j \leq N.$$

*Proof.* Fix any $T \in \mathbb{N}_{\geq 0}$. Since $\|P^{(t)}\|_{\infty} \leq 1$ for any $P^{(t)} \in \mathcal{P}_{\mathcal{G},\epsilon}$, it follows that $\|P^{(t_0+T:t_0)}\|_{\infty} \leq 1$ and hence $P_{ij}^{(t_0+T:t_0)} \leq 1$, for all $1 \leq i,j \leq N$.

To show the lower bound, without loss of generality, we will show that there exist $T \in \mathbb{N}$ and $c > 0$ such that

$$P_{ij}^{(T:0)} \geq c, \forall 1 \leq i,j \leq N.$$

Since each $P^{(t)}$ has the same connectivity pattern as the original graph $\mathcal{G}$, it follows from the assumption **A1** that there exists $T \in \mathbb{N}$ such that $P^{(T:0)}$ is a positive matrix, following a similar argument as the one for Proposition 1.7 in [23]: For each pair of nodes $i, j$, since we assume that the graph $\mathcal{G}$ is connected, there exists $r(i,j)$ such that $P_{ij}^{(r(i,j):0)} > 0$. on the other hand, since we also assume each node has a self-loop, $P_{ii}^{(t:0)} > 0$ for all $t \geq 0$ and hence for $t \geq r(i,j)$,

$$P_{ij}^{(t:0)} \geq P_{ii}^{(t-r(i,j))} P_{ij}^{(r(i,j):0)} > 0.$$

For $t \geq t(i) := \max_{j \in \mathcal{G}} r(i,j)$, we have $P_{ij}^{(t:0)} > 0$ for all node $j$ in $\mathcal{G}$. Finally, if $t \geq T := \max_{i \in \mathcal{G}} t(i)$, then $P_{ij}^{(t:0)} > 0$ for all pairs of nodes $i, j$ in $\mathcal{G}$. Notice that $P_{ij}^{(T:0)}$ is a weighted sum of walks of length $T$ between nodes $i$ and $j$, and hence $P_{ij}^{(T:0)} > 0$ if and only if there exists a walk of length $T$ between nodes $i$ and $j$. Since for all $t \in \mathbb{N}_{\geq 0}$, $P_{ij}^{(t)} \geq \epsilon$ if $(i,j) \in E(\mathcal{G})$, we conclude that $P_{ij}^{(T:0)} \geq \epsilon^T := c$. $\qquad\square$

## E.2 Proof of Lemma 3

Given the sequence $\{P^{(t)}\}_{t=0}^{\infty}$, we use $T \in \mathbb{N}$ from Lemma 9 and define

$$\bar{P}^{(k)} := P^{((k+1)T:kT)}.$$

Then $\{P^{(t)}\}_{t=0}^{\infty}$ is ergodic if and only if $\{\bar{P}^{(k)}\}_{k=0}^{\infty}$ is ergodic. Notice that by Lemma 9, for all $k \in \mathbb{N}_{\geq 0}$, there exists $c > 0$ such that $c \leq \bar{P}_{ij}^{(k)} \leq 1, \forall 1 \leq i,j \leq N$. Then Lemma 3 is a direct consequence of Lemma 8.

# F Proof of Lemma 5

## F.1 Notations and auxiliary results

Consider a sequence $\{D^{(t)}P^{(t)}\}_{t=0}^{\infty}$ in $\mathcal{M}_{\mathcal{G},\epsilon}$. For $t_0 \leq t_1$, define

$$Q_{t_0,t_1} := D^{(t_1)}P^{(t_1)}...D^{(t_0)}P^{(t_0)}$$

and

$$\delta_t = \|D^{(t)} - I_N\|_{\infty},$$

where $I_N$ denotes the $N \times N$ identity matrix. It is also useful to define

$$\begin{aligned}\hat{Q}_{t_0,t_1} &:= P^{(t_1)}Q_{t_0,t_1-1} \\ &:= P^{(t_1)}D^{(t_1-1)}P^{(t_1-1)}...D^{(t_0)}P^{(t_0)}.\end{aligned}$$

We start by proving the following key lemma, which states that long products of matrices in $\mathcal{M}_{\mathcal{G},\epsilon}$ eventually become a contraction in $\infty$-norm.

**Lemma 10.** *There exist $0 < c < 1$ and $T \in \mathbb{N}$ such that for all $t_0 \leq t_1$,*

$$\|\hat{Q}_{t_0,t_1+T}\|_{\infty} \leq (1 - c\delta_{t_1})\|\hat{Q}_{t_0,t_1}\|_{\infty}.$$

*Proof.* First observe that for every $T \geq 0$,

$$\begin{aligned}\|\hat{Q}_{t_0,t_1+T}\|_{\infty} &\leq \|P^{(t_1+T)}D^{(t_1+T-1)}P^{(t_1+T-1)}...D^{(t_1+1)}P^{(t_1+1)}D^{(t_1)}\|_{\infty}\|\hat{Q}_{t_0,t_1}\|_{\infty} \\ &\leq \|P^{(t_1+T)}P^{(t_1+T-1)}...P^{(t_1+1)}D^{(t_1)}\|_{\infty}\|\hat{Q}_{t_0,t_1}\|_{\infty},\end{aligned}$$

where the second inequality is based on the following element-wise inequality:

$$P^{(t_1+T)}D^{(t_1+T-1)}P^{(t_1+T-1)}...D^{(t_1+1)}P^{(t_1+1)} \leq_{\text{ew}} P^{(t_1+T)}P^{(t_1+T-1)}...P^{(t_1+1)}.$$

By Lemma 9, there exist $T \in \mathbb{N}$ and $0 < c < 1$ such that

$$(P^{(t_1+T)}...P^{(t_1+1)})_{ij} \geq c, \forall 1 \leq i, j \leq N.$$

Since the matrix product $P^{(t_1+T)}P^{(t_1+T-1)}...P^{(t_1+1)}$ is row-stochastic, multiplying it with the diagonal matrix $D^{(t_1)}$ from right decreases the row sums by at least $c(1 - D_{\min}^{(t_1)}) = c\delta_{t_1}$, where $D_{\min}^{(t_1)}$ here denotes the smallest diagonal entry of the diagonal matrix $D^{(t_1)}$. Hence,

$$\|P^{(t_1+T)}P^{(t_1+T-1)}...P^{(t_1+1)}D^{(t_1)}\|_{\infty} \leq 1 - c\delta_{t_1}.$$

$\square$

## F.2 Proof of Lemma 4

Now define $\beta_k := \prod_{t=0}^{k}(1 - c\delta_t)$ and let $\beta := \lim_{k\to\infty} \beta_k$. Note that $\beta$ is well-defined because the partial product is non-increasing and bounded from below. Then we present the following result, which is stated as Lemma 4 in the main paper and from which the ergodicity of any sequence in $\mathcal{M}_{\mathcal{G},\epsilon}$ is an immediate result.

**Lemma 4.** Let $\beta_k := \prod_{t=0}^{k}(1 - c\delta_t)$ and $\beta := \lim_{k\to\infty} \beta_k$.

1. If $\beta = 0$, then $\lim_{k\to\infty} Q_{0,k} = 0$;

2. If $\beta > 0$, then $\lim_{k\to\infty} BQ_{0,k} = 0$.

*Proof.* We will prove the two cases separately.

**[Case $\beta = 0$]** We will show that $\beta = 0$ implies $\lim_{k\to\infty} \|\hat{Q}_{0,k}\|_\infty = 0$, and as a result, $\lim_{k\to\infty} \|Q_{0,k}\|_\infty = 0$. For $0 \le j \le T - 1$, let us define

$$\beta^j := \prod_{k=0}^{\infty} (1 - \delta_{j+kT}).$$

Then by Lemma 10, we get that

$$\lim_{k\to\infty} \|\hat{Q}_{0,kT}\|_\infty \le \beta^j \|\hat{Q}_{0,j}\|_\infty.$$

By construction, $\beta = \Pi_{j=0}^{T-1} \beta^j$. Hence, if $\beta = 0$ then $\beta^{j_0} = 0$ for some $0 \le j_0 \le T - 1$, which yields $\lim_{k\to\infty} \|\hat{Q}_{0,k}\|_\infty = 0$. Consequently, $\lim_{k\to\infty} \|Q_{0,k}\|_\infty = 0$ implies that $\lim_{k\to\infty} Q_{0,k} = 0$.

**[Case $\beta > 0$]** First observe that if $\beta > 0$, then $\forall 0 < \eta < 1$, there exist $m \in \mathbb{N}_{\ge 0}$ such that

$$\prod_{t=m}^{\infty} (1 - c\delta_t) > 1 - \eta. \tag{9}$$

Using $1 - x \le e^{-x}$ for all $x \in \mathbb{R}$, we deduce

$$\prod_{t=m}^{\infty} e^{-c\delta_t} > 1 - \eta.$$

It also follows from (9) that $1 - c\delta_t > 1 - \eta$, or equivalently $\delta_t < \frac{\eta}{c}$ for $t \ge m$. Choosing $\eta < \frac{c}{2}$ thus ensures that $\delta_t < \frac{1}{2}$ for $t \ge m$. Putting this together with the fact that, there exists[4] $b > 0$ such that $1 - x \ge e^{-bx}$ for all $x \in [0, \frac{1}{2}]$, we obtain

$$\prod_{t=m}^{\infty} (1 - \delta_t) \ge \prod_{t=m}^{\infty} e^{-b\delta_t} > (1 - \eta)^{\frac{b}{c}} := 1 - \eta'. \tag{10}$$

Define the product of row-stochastic matrices $P^{(M:m)} := P^{(M)} \ldots P^{(m)}$. It is easy to verify the following element-wise inequality:

$$\left( \prod_{t=m}^{M} (1 - \delta_t) \right) P^{(M:m)} \le_{\text{ew}} Q_{m,M} \le_{\text{ew}} P^{(M:m)},$$

which together with (10) leads to

$$(1 - \eta') P^{(M:m)} \le_{\text{ew}} Q_{m,M} \le_{\text{ew}} P^{(M:m)}. \tag{11}$$

Therefore,

$$\begin{aligned}
\|BQ_{m,M}\|_\infty &= \|B(Q_{m,M} - P^{(M:m)}) + BP^{(M:m)}\|_\infty \\
&\le \|B(Q_{m,M} - P^{(M:m)})\|_\infty + \|BP^{(M:m)}\|_\infty \\
&= \|B(Q_{m,M} - P^{(M:m)})\|_\infty \\
&\le \|B\|_\infty \|Q_{m,M} - P^{(M:m)}\|_\infty \\
&\le \eta' \|B\|_\infty \\
&\le \eta' \sqrt{N},
\end{aligned}$$

where the last inequality is due to the fact that $\|B\|_2 = 1$. By definition, $Q_{0,M} = Q_{m,M} Q_{0,m-1}$, and hence

$$\|BQ_{0,M}\|_\infty \le \|BQ_{m,M}\|_\infty \|Q_{0,m-1}\|_\infty \le \|BQ_{m,M}\|_\infty \le \eta' \sqrt{N}. \tag{12}$$

The above inequality (12) holds when taking $M \to \infty$. Then taking $\eta \to 0$ implies $\eta' \to 0$ and together with (12), we conclude that

$$\lim_{M\to\infty} \|BQ_{0,M}\|_\infty = 0,$$

---

[4]Choose, e.g., $b = 2 \log 2$.

and therefore,

$$\lim_{M\to\infty} BQ_{0,M} = 0.$$

$\square$

### F.3 Proof of Lemma 5

Notice that both cases $\beta = 0$ and $\beta > 0$ in Lemma 4 imply the ergodicity of $\{D^{(t)}P^{(t)}\}_{t=0}^{\infty}$. Hence the statement is a direct corollary of Lemma 4.

## G Proof of Lemma 6

In order to show that $\mathrm{JSR}(\tilde{\mathcal{P}}_{\mathcal{G},\epsilon}) < 1$, we start by making the following observation.

**Lemma 11.** *A sequence $\{P^{(n)}\}_{n=0}^{\infty}$ is ergodic if and only if $\prod_{n=0}^{t} \tilde{P}^{(n)}$ converges to the zero matrix.*

*Proof.* For any $t \in \mathbb{N}_{\geq 0}$, it follows from the third property of the orthogonal projection $B$ (see, Page 6 of the main paper) that

$$B\prod_{n=0}^{t} P^{(n)} = \prod_{n=0}^{t} \tilde{M}^{(n)}P.$$

Hence

$$\{P^{(n)}\}_{n=0}^{\infty} \text{ is ergodic} \iff \lim_{t\to\infty} B\prod_{n=0}^{t} P^{(n)} = 0$$

$$\iff \lim_{t\to\infty} \prod_{n=0}^{t} \tilde{P}^{(n)}B = 0$$

$$\iff \lim_{t\to\infty} \prod_{n=0}^{t} \tilde{P}^{(n)} = 0.$$

$\square$

Next, we utilize the following result, as a means to ensure a joint spectral radius strictly less than 1 for a bounded set of matrices.

**Lemma 12** (Proposition 3.2 in [35]). *For any bounded set of matrices $\mathcal{M}$, $\mathrm{JSR}(\mathcal{M}) < 1$ if and only if for any sequence $\{M^{(n)}\}_{n=0}^{\infty}$ in $\mathcal{M}$, $\prod_{n=0}^{t} M^{(n)}$ converges to the zero matrix.*

Here, "bounded" means that there exists an upper bound on the norms of the matrices in the set. Note that $\mathcal{P}_{\mathcal{G},\epsilon}$ is bounded because $\|P\|_{\infty} = 1$, for all $P \in \mathcal{M}_{\mathcal{G},\epsilon}$. To show that $\tilde{\mathcal{P}}_{\mathcal{G},\epsilon}$ is also bounded, let $\tilde{P} \in \tilde{\mathcal{P}}_{\mathcal{G},\epsilon}$, then by definition, we have

$$\tilde{P}B = BP, P \in \mathcal{M}_{\mathcal{G},\epsilon} \Rightarrow \tilde{P} = BPB^{T},$$

since $BB^{T} = I_{N-1}$. As a result,

$$\|\tilde{P}\|_{2} = \|BPB^{T}\|_{2} \leq \|P\|_{2} \leq \sqrt{N},$$

where the first inequality is due to $\|B\|_{2} = \|B^{\top}\|_{2} = 1$, and the second ineuality follows from $\|P\|_{\infty} = 1$.

Combining Lemma 5, Lemma 11 and Lemma 12, we conclude that $\mathrm{JSR}(\tilde{\mathcal{P}}_{\mathcal{G},\epsilon}) < 1$.

## H Proof of Lemma 7

Note that any sequence $\{M^{(n)}\}_{n=0}^{\infty}$ in $\mathcal{M}_{\mathcal{G},\epsilon,\delta}$ satisfies $\beta = 0$, where $\beta$ is defined in Lemma 4. This implies that for any sequence $\{M^{(n)}\}_{n=0}^{\infty}$ in $\mathcal{M}_{\mathcal{G},\epsilon,\delta}$, we have

$$\lim_{t\to\infty} \prod_{n=0}^{t} M^{(n)} = 0.$$

Since $\|M\|_\infty \le \delta$ for all $M \in \mathcal{M}_{\mathcal{G},\epsilon,\delta}$, again by Lemma 12, we conclude that $\mathrm{JSR}(\mathcal{M}_{\mathcal{G},\epsilon,\delta}) < 1$.

# I  Proof of Theorem 1

Recall the formulation of $X_{\cdot i}^{(t+1)}$ in (7):

$$X_{\cdot i}^{(t+1)} = \sigma(P^{(t)}(X^{(t)}W^{(t)})_{\cdot i}) = \sum_{j_{t+1}=i,\,(j_t,\ldots,j_0)\in[d]^{t+1}} \left(\prod_{k=0}^{t} W_{j_k j_{k+1}}^{(k)}\right) D_{j_{t+1}}^{(t)} P^{(t)}...D_{j_1}^{(0)} P^{(0)} X_{\cdot j_0}^{(0)}.$$

Then the first part of the theorem directly follows from Lemma 5.

To derive the exponential convergence rate, consider the linear and nonlinear cases separately:

## I.1  Bounds for the two cases

**[Case: linear]**  In the linear case where all $D = I_N$, it follows from Lemma 6 that

$$
\begin{aligned}
\|BX_{\cdot i}^{(t+1)}\|_2 &= \left\|\sum_{j_{t+1}=i,\,(j_t,\ldots,j_0)\in[d]^{t+1}} \left(\prod_{k=0}^{t} W_{j_k j_{k+1}}^{(k)}\right) BP^{(t)}...P^{(0)} X_{\cdot j_0}^{(0)}\right\|_2 \\
&\le \sum_{j_{t+1}=i,\,(j_t,\ldots,j_0)\in[d]^{t+1}} \left(\prod_{k=0}^{t} \left|W_{j_k j_{k+1}}^{(k)}\right|\right) \left\|BP^{(t)}...P^{(0)} X_{\cdot j_0}^{(0)}\right\|_2 \\
&= \sum_{j_{t+1}=i,\,(j_t,\ldots,j_0)\in[d]^{t+1}} \left(\prod_{k=0}^{t} \left|W_{j_k j_{k+1}}^{(k)}\right|\right) \left\|\tilde{P}^{(t)}...\tilde{P}^{(0)} BX_{\cdot j_0}^{(0)}\right\|_2 \\
&\le \sum_{j_{t+1}=i,\,(j_t,\ldots,j_0)\in[d]^{t+1}} \left(\prod_{k=0}^{t} \left|W_{j_k j_{k+1}}^{(k)}\right|\right) Cq^{t+1} \left\|BX_{\cdot j_0}^{(0)}\right\|_2 \\
&\le C'q^{t+1} \left(\sum_{j_{t+1}=i,\,(j_t,\ldots,j_0)\in[d]^{t+1}} \left(\prod_{k=0}^{t} \left|W_{j_k j_{k+1}}^{(k)}\right|\right)\right) \\
&= C'q^{t+1}\|(|W^{(0)}|...|W^{(t)}|)_{\cdot i}\|_1 \,,
\end{aligned}
\tag{13}
$$

where $C' = C\max_{j\in[d]}\|BX_{\cdot j}^{(0)}\|_2$ and $\|\cdot\|_1$ denotes the 1-norm. Specifically, the first inequality follows from the triangle inequality, and the second inequality is due to the property of the joint spectral radius in (8), where $\mathrm{JSR}(\tilde{\mathcal{P}}_{\mathcal{G},\epsilon}) < q < 1$.

**[Case: nonlinear]**  Consider $T \in \mathbb{N}$ defined in Lemma 10, where there exists $a_1 \in \mathbb{N}_{\ge 0}$ and $a_2 \in \{0\} \cup [K-1]$ such that $T = a_1 K + a_2$. Given the condition $(\star)$, and Lemma 10, there exists $0 < c < 1$ such that for all $m \in \mathbb{N}_{\ge 0}$,
$$\|\hat{Q}_{mK+n_m,mK+n_m+T}\|_\infty \le (1-c\delta')\,,$$
where $\delta' = 1 - \delta$. Note that since for all $n_m, m \in \mathbb{N}_{\ge 0}$, $a_2 + n_m \le 2K$, we get that
$$n_m + T \le (a_1+2)K\,.$$

Since for all $m \in \mathbb{N}_{\ge 0}$,

$$\|\hat{Q}_{mK+n_m:(m+a_1+2)K+n_{m+a_1+2}}\|_\infty \le (1-c\delta')\,,$$

it implies that for all $n \in \mathbb{N}_{\ge 0}$,

$$
\begin{aligned}
\|Q_{0:n(a_1+2)K}\|_\infty \le (1-c\delta')^n &= \left((1-c\delta')^{\frac{n}{n(a_1+2)K}}\right)^{n(a_1+2)K} \\
&= \left((1-c\delta')^{\frac{1}{(a_1+2)K}}\right)^{n(a_1+2)K}.
\end{aligned}
$$

Denote $\underline{q} := (1 - c\delta')^{\frac{1}{(a_1+2)K}}$. By the equivalence of norms, we get that for all $n \in \mathbb{N}_{\geq 0}$,

$$\|Q_{0:n(a_1+2)K}\|_2 \leq \sqrt{N}\underline{q}^{n(a_1+2)K} .$$

Then for any $j \in \{0\} \cup [(a_1+2)K - 1]$,

$$\|Q_{0:n(a_1+2)K+j}\|_2 \leq \sqrt{N}\underline{q}^{-j}\underline{q}^{n(a_1+2)K+j} \leq C\underline{q}^{n(a_1+2)K+j}$$

where $C := \sqrt{N}\underline{q}^{-(a_1+2)K+1}$. Rewriting the indices, we conclude that

$$\|Q_{0:t}\|_2 \leq C\underline{q}^t , \forall t \geq 0 .$$

The above bound implies that for any $q$ satisfies $\underline{q} \leq q < 1$,

$$
\begin{aligned}
\|BX_{\cdot i}^{(t+1)}\|_2 &= \left\| \sum_{j_{t+1}=i,\,(j_t,\ldots,j_0)\in[d]^{t+1}} \left( \prod_{k=0}^{t} W_{j_k j_{k+1}}^{(k)} \right) BD_{j_{t+1}}^{(t)} P^{(t)} \ldots D_{j_1}^{(0)} P^{(0)} X_{\cdot j_0}^{(0)} \right\|_2 \\
&\leq \sum_{j_{t+1}=i,\,(j_t,\ldots,j_0)\in[d]^{t+1}} \left( \prod_{k=0}^{t} \left| W_{j_k j_{k+1}}^{(k)} \right| \right) \left\| BD_{j_{t+1}}^{(t)} P^{(t)} \ldots D_{j_1}^{(0)} P^{(0)} X_{\cdot j_0}^{(0)} \right\|_2 \\
&\leq \sum_{j_{t+1}=i,\,(j_t,\ldots,j_0)\in[d]^{t+1}} \left( \prod_{k=0}^{t} \left| W_{j_k j_{k+1}}^{(k)} \right| \right) \left\| D_{j_{t+1}}^{(t)} P^{(t)} \ldots D_{j_1}^{(0)} P^{(0)} \right\| \left\| X_{\cdot j_0}^{(0)} \right\|_2 \\
&\leq C'q^{t+1} \left( \sum_{j_{t+1}=i,\,(j_t,\ldots,j_0)\in[d]^{t+1}} \left( \prod_{k=0}^{t} \left| W_{j_k j_{k+1}}^{(k)} \right| \right) \right) \\
&= C'q^{t+1} \| (|W^{(0)}|\ldots|W^{(t)}|)_{\cdot i} \|_1 ,
\end{aligned}
\tag{14}
$$

where again, $C' = C\max_{j\in[d]}\|BX_{\cdot j}^{(0)}\|_2$.

## I.2 Proof of the exponential convergence

Based on the inequality extablished for both linear and nonlinear case in (13) and (14), we derive the rest of the proof. Since $\|Bx\|_2 = \|x\|_2$ if $x^\top \mathbf{1} = 0$ for $x \in \mathbb{R}^N$, we also have that if $X^\top \mathbf{1} = 0$ for $X \in \mathbb{R}^{N\times d}$, then

$$\|BX\|_F = \|X\|_F ,$$

using which we obtain that

$$
\begin{aligned}
\mu(X^{(t+1)}) = \|X^{(t+1)} - \mathbf{1}\gamma_{X^{(t+1)}}\|_F = \|BX^{(t+1)}\|_F &= \sqrt{\sum_{i=1}^{d} \|BX_{\cdot i}^{(t+1)}\|_2^2} \\
&\leq C'q^{t+1} \sqrt{\sum_{i=1}^{d} \|(|W^{(0)}|\ldots|W^{(t)}|)_{\cdot i}\|_1^2} \\
&\leq C'q^{t+1} \sqrt{\left( \sum_{i=1}^{d} \||(|W^{(0)}|\ldots|W^{(t)}|)_{\cdot i}\|_1 \right)^2} \\
&= C'q^{t+1} \||(|W^{(0)}|\ldots|W^{(t)}|\|_{1,1} ,
\end{aligned}
$$

where $\|\cdot\|_{1,1}$ denotes the matrix $(1,1)$-norm (recall from Section A.2 that for a matrix $M \in \mathbb{R}^{m\times n}$, we have $\|M\|_{1,1} = \sum_{i=1}^{m}\sum_{j=1}^{n}|M_{ij}|$). The assumption **A3** implies that there exists $C''$ such that for all $t \in \mathbb{N}_{\geq 0}$,

$$\|(|W^{(0)}|\ldots|W^{(t)}|)\|_{1,1} \leq C''d^2 .$$

Thus we conclude that there exists $C_1$ such that for all $t \in \mathbb{N}_{\geq 0}$,

$$\mu(X^{(t)}) \leq C_1 q^t \,.$$

**Remark 3.** *Similar to the linear case, one can also use Lemma 7 to establish exponential rate for oversmoothing when dealing with nonlinearities for which Assumption 4 holds in the strict sense, i.e. $0 \leq \frac{\sigma(x)}{x} < 1$ (e.g., GELU and SiLU nonlinearities). Here, we presented an alternative proof requiring weaker conditions, making the result directly applicable to nonlinearities such as ReLU and Leaky ReLU.*

## J  Proof of Proposition 2

Since $D_{\mathrm{deg}}^{-1}A$ is similar to $D_{\mathrm{deg}}^{-1/2}AD_{\mathrm{deg}}^{-1/2}$, they have the same spectrum. For $D_{\mathrm{deg}}^{-1}A$, the smallest nonzero entry has value $1/d_{\max}$, where $d_{\max}$ is the maximum node degree in $\mathcal{G}$. On the other hand, it follows from the definition of $\mathcal{P}_{\mathcal{G},\epsilon}$ that

$$\epsilon d_{\max} \leq 1 \,.$$

Therefore, $\epsilon \leq 1/d_{\max}$ and thus $D_{\mathrm{deg}}^{-1}A \in \mathcal{P}_{\mathcal{G},\epsilon}$.

We proceed by proving the following result.

**Lemma 13.** *For any $M$ in $\mathcal{M}$, the spectral radius of $M$ denoted by $\rho(M)$, satisfies*

$$\rho(M) \leq \mathrm{JSR}(\mathcal{M}) \,.$$

*Proof.* Gelfand's formula states that $\rho(M) = \lim_{k \to \infty} \|M^k\|^{\frac{1}{k}}$, where the quantity is independent of the norm used [22]. Then comparing with the definition of the joint spectral radius, we can immediately conclude the statement. $\square$

Let $B(D_{\mathrm{deg}}^{-1}A) = \tilde{P}B$. By definition, $\tilde{P} \in \tilde{\mathcal{P}}_{\mathcal{G},\epsilon}$ since $D_{\mathrm{deg}}^{-1}A \in \mathcal{P}_{\mathcal{G},\epsilon}$ as shown before the lemma. Moreover, the spectrum of $\tilde{P}$ is the spectrum of $D_{\mathrm{deg}}^{-1}A$ after reducing the multiplicity of eigenvalue 1 by one. Under the assumption **A1**, the eigenvalue 1 of $D_{\mathrm{deg}}^{-1}A$ has multiplicity 1, and hence $\rho(\tilde{P}) = \lambda$, where $\lambda$ is the second largest eigenvalue of $D_{\mathrm{deg}}^{-1}A$. Putting this together with Lemma 13, we conclude that

$$\lambda \leq \mathrm{JSR}(\tilde{\mathcal{P}}_{\mathcal{G},\epsilon})$$

as desired.

## K  Numerical Experiments

Here we provide more details on the numerical experiments presented in Section 5. All models were implemented with PyTorch [27] and PyTorch Geometric [10].

**Datasets**

- We used `torch_geometric.datasets.planetoid` provided in PyTorch Geometric for the three homophilic datasets: Cora, CiteSeer, and PubMed with their default training and test splits.
- We used `torch_geometric.datasets.WebKB` provided in PyTorch Geometric for the three heterophilic datasets: Cornell, Texas, and Wisconsin with their default training and test splits.
- Dataset summary statistics are presented in Table 1.

**Model details**

- For GAT, we consider the architecture proposed in Veličković et al. [38] with each attentional layer sharing the parameter $a$ in $\mathrm{LeakyReLU}(a^\top[W^\top X_i \| W^\top X_j]), a \in \mathbb{R}^{2d'}$ to compute the attention scores.

| Dataset | Type | #Nodes | %Nodes in LCC |
|---------|------|--------|---------------|
| Cora | | 2,708 | 91.8 % |
| CiteSeer | homophilic | 3,327 | 63.7 % |
| PubMed | | 19,717 | 100 % |
| Cornell | | 183 | 100 % |
| Texas | heterophilic | 183 | 100 % |
| Wisconsin | | 251 | 100 % |
| Flickr | large-scale | 89,250 | 100 % |

Table 1: Dataset summary statistics. LCC: Largest connected component.

- For GCN, we consider the standard random walk graph convolution $D_{\mathrm{deg}}^{-1}A$. That is, the update rule of each graph convolutional layer can be written as

$$X' = D_{\mathrm{deg}}^{-1}AXW\,,$$

where $X$ and $X'$ are the input and output node representations, respectively, and $W$ is the shared learnable weight matrix in the layer.

**Compute**   We trained all of our models on a Telsa V100 GPU.

**Training details**   In all experiments, we used the Adam optimizer using a learning rate of $0.00001$ and $0.0005$ weight decay and trained for $1000$ epoch

**Results on large-scale dataset**   In addition to the numerical results presented in Section 5, we also conducted the same experiment on a large-scale dataset Flickr [46] using `torch_geometric.datasets.Flickr`. Figure 2 visualizes the results.

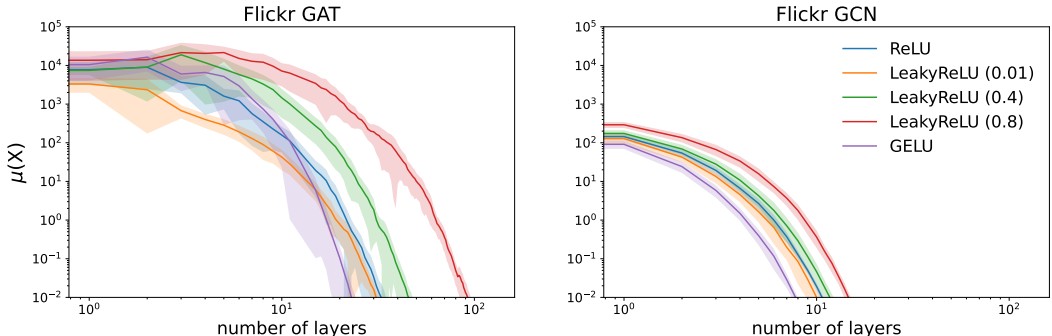

Figure 2: Evolution of $\mu(X^{(t)})$ (in log-log scale) on the largest connected component of the large-scale benchmark dataset: Flickr.

