# OpenReview forum: "Demystifying Oversmoothing in Attention-Based Graph Neural Networks"
_NeurIPS.cc/2023/Conference — NeurIPS 2023 spotlight_

### Official Review · Reviewer_UfCo · 2023-06-09

**Soundness:** 4 excellent
**Presentation:** 3 good
**Contribution:** 4 excellent
**Rating:** 7
**Confidence:** 4

**Summary:**

The authors provide theoretical proofs on the oversmoothing problem in GCNs and GATs through Ergodicity analysis of infinite product of matrices. The authors provide a bound in convergence of oversmoothing problem and provides suggestions on choices in activation function to relief oversmoothing problem.

**Strengths:**

- Approaching oversmoothing problem with Ergodicity is fairly new
- Implications on choices of activation functions are straight forward
- The article is very easy to follow

**Weaknesses:**

- It would be better if the authors could add a experiment with a plot directly showing JSR

**Questions:**

- If we break the assumption A4 by using a $\sigma(x)/x >1$, or $\sigma(0)\neq 0$, does GNNs stop having oversmoothing issues or will there be other problems?

---

> ### Author Rebuttal · Authors · 2023-08-10
>
>
> Thank you for your constructive feedback and positive assessment of our work. Below, we provide individual responses to your questions.
>
> **Q1: It would be better if the authors could add a experiment with a plot directly showing JSR.**
>
> Thank you for the comment. As mentioned in the paper (line 305-306),  computing or approximating the joint spectral radius for a given set of matrices is in fact an NP-hard problem in general [1]. However, using the proof techniques we develop in the paper, we show that the joint spectral radius of the set of matrices $\tilde{\mathcal{M}}\_{\mathcal{G},\epsilon}$ is strictly less than 1, from which we then derive our oversmoothing results.
>
> **Q2: If we break the assumption A4 by using $\sigma(x)/x > 1$ or $\sigma(0) \neq 0$, does GNNs stop having oversmoothing issues or will there be other problems?**
>
> This is a great follow-up research question that we are currently investigating.
>
> - We conjecture that choosing a nonlinearity with $\sigma(x)/x>1$ may enable us to break the invariance of the graph structure across layers in the limit (which was the key cause for oversmoothing in our analysis). The idea is that such a $\sigma(\cdot)$ may lead to unbounded raw attention scores in the limit. When normalized by softmax, this can result in zero attention coefficient, which is equivalent to edge-dropping.
>
> - There will be both cons and pros for such a choice: while having several disconnected components can potentially lead to a much richer structure for the limit points of the trajectories, allowing for large outputs may lead to optimization issues in training.
>
> We appreciate your questions and comments very much. Please let us know for any further questions.
>
> ---
> **References**
>
> [1] John Tsitsiklis and Vincent Blondel. The Lyapunov exponent and joint spectral radius of pairs of matrices are hard – when not impossible – to compute and to approximate. Mathematics of Control, Signals and Systems, 1997.

---

### Official Review · Reviewer_DswA · 2023-06-19

**Soundness:** 4 excellent
**Presentation:** 3 good
**Contribution:** 3 good
**Rating:** 7
**Confidence:** 3

**Summary:**

The objective of this paper is to examine the occurrence of over-smoothing phenomena in attention-based Graph Neural Networks (GNN). Over-smoothing refers to a situation where deepening the network results in similar node representations. The researchers demonstrate the presence of over-smoothing in attention-based GNNs using a versatile methodology that can be applied to various GNN types. They establish this proof by treating the network as a nonlinear and time-varying dynamical system. Moreover, the authors support their assertions with empirical evidence.

**Strengths:**

* The paper exhibits excellent writing quality, as it effectively communicates the author's intentions even when dealing with complex mathematical concepts. The clarity of the paper makes it easily comprehensible for readers.

* The theoretical claims are clear and the proofs are valid.

* The authors propose a generic approach to examine over-smoothing in GNNs with non-restrictive conditions. While their main focus is on attention-based models, the results include a wider range of models.


**Weaknesses:**

* The main shortcoming of the paper is the lack of suggestions for solving the over-smoothing problem. [1] suggests scaling to avoid the phenomena, are they any other ways to alleviate the problem?

[1] Oono et al. Graph Neural Networks Exponentially Lose Expressive Power for Node Classification, ICLR (2020)


**Questions:**

* Is it possible to extend the over-smoothing to any node similarity measure as defined in Definition 1 ?
* Is it possible to directly expand the results from the paper to transformer-based GNNs? What is possible to say about the existence of over-smoothing in that case?


**Limitations:**

There is no evidence of ethical problems with this submission.

---

> ### Author Rebuttal · Authors · 2023-08-10
>
> We greatly appreciate your positive assessment and insightful comments, which have helped strengthen our work. Below, we provide individual responses to your questions.
>
> **Q1: The main shortcoming of the paper is the lack of suggestions for solving the over-smoothing problem. [1] suggests scaling to avoid the phenomena, are there any other ways to alleviate the problem?**
>
> Thank you for the comment. As our discussion following Theorem 1 (lines 297-303) reveals, a pivotal factor contributing to oversmoothing is the invariance of the graph structure across different layers. This supports the empirical approaches to mitigate oversmoothing in the existing literature such as edge-dropping or graph-rewiring (citations provided in the paper), as they modify the connectivity pattern of the graph.
>
> **Q2: Is it possible to extend the over-smoothing to any node similarity measure as defined in Definition 1?**
>
> This is a very good question and helps us strengthen our theoretical results. In fact, our analysis directly applies to any Lipschitz node similarity measure $\mu$. To see this, observe that for a node similarity measure $\mu$ with a Lipschitz constant $C$, we have $\mu(X) =  \|\mu(X) - \mu(1\_{\gamma\_X})\| \leq C ||X - 1\_{\gamma\_X}||_F$, where $||X - 1\_{\gamma\_X}||_F$ is the node similarity measure considered in this work.
>
> **Q3: Is it possible to directly expand the results from the paper to transformer-based GNNs? What is possible to say about the existence of over-smoothing in that case?**
>
> This is again a great question. In fact, one can view the self-attention dynamics in transformers as a special class of attention-based GNNs operating on complete graphs. Then our results would directly apply to transformer-based GNNs under similar assumptions to A2-A4.
>
> We appreciate your questions and comments very much. Please let us know if there are any further questions.
>
> ---
> **References**
>
> [1] Oono and Suzuki. Graph Neural Networks Exponentially Lose Expressive Power for Node Classification. In ICLR 2020.

---

### Official Review · Reviewer_Ac2q · 2023-07-06

**Soundness:** 3 good
**Presentation:** 3 good
**Contribution:** 3 good
**Rating:** 6
**Confidence:** 5

**Summary:**

This paper provides a definitive answer to the question that whether the graph attention mechanism can mitigate over-smoothing through a rigorous mathematical analysis. The theoretical results establish that the graph attention mechanism cannot prevent over-smoothing and loses expressive power exponentially.

**Strengths:**

S1. Exploring theoretical results on the performance of attention-based GNNs on over-smoothing problems is meaningful and challenging.

S2. The theoretical results obtained in this paper can provide some instructive and valuable information on over-smoothing research.

**Weaknesses:**

Reliability
1. The reliability of the motivation can be enhanced by adding more literature that discusses the controversy surrounding whether attention-based GNNs alleviate the over-smoothing problem.

Significance
1. Only homophily graphs are considered in the paper. In addition, heterophily datasets also should be considered to verify the overall performance of GNN as pointed out by recent publications [1,2].
2. In addition, experiments on larger datasets will strengthen the significance of the contributions.

[1]Pei, H., Wei, B., Chang, K. C. C., Lei, Y., & Yang, B. (2020). Geom-GCN: Geometric graph convolutional networks. In ICLR.

[2]Lim, D., Hohne, F., Li, X., Huang, S. L., Gupta, V., Bhalerao, O., & Lim, S. N. (2021). Large scale learning on non-homophilous graphs: New benchmarks and strong simple methods. In NeurIPS, 34, 20887-20902.

**Questions:**

1.Missing a number of reliable literature support the claim of attention-based GNNs' ability to alleviate the over-smoothing problem, and most of the works listed in this paper suggest that attention-based GNNs suffer from over-smoothing. This is insufficient to support the author's argument that there is controversial whether the graph attention mechanism can mitigate over-smoothing.

2.What criteria were used to select the negative slope values in the experiment?

3.How sensitive are the hyperparameters (e.g. negative slope values) to the performance?

---

> ### Author Rebuttal · Authors · 2023-08-10
>
> We thank the reviewer for taking the time to review our paper and providing constructive feedback. In line with the reviewer’s suggestion, we have performed additional numerical experiments, validating our theoretical findings on three heterophily datasets (Cornell, Texas and Wisconsin [1]) and a large-scale dataset  (Flickr [2]). The results are provided in the rebuttal pdf file. In what follows, we provide detailed responses to the comments raised by the reviewer.
>
> **Q1: Missing a number of reliable literature support the claim of attention-based GNNs' ability to alleviate the over-smoothing problem, and most of the works listed in this paper suggest that attention-based GNNs suffer from over-smoothing. This is insufficient to support the author's argument that there is controversial whether the graph attention mechanism can mitigate over-smoothing. The reliability of the motivation can be enhanced by adding more literature that discusses the controversy surrounding whether attention-based GNNs alleviate the over-smoothing problem.**
>
> Thank you for the comment. We would like to clarify that by "controversy" here, we mean it is still an “open question” (especially from a *theoretical* perspective) whether graph attention mechanisms can mitigate oversmoothing [3,4]. For example, Min et al. [5] directly claim that “GAT network typically alleviates oversmoothing using graph attention mechanisms,” while oversmoothing in attention-based GNNs has been observed empirically in some other works [6]. As also pointed out by the reviewer, “exploring theoretical results on the performance of attention-based GNNs on oversmoothing problems is meaningful and challenging.” Our paper fills the gap in the existing theoretical literature by providing a definitive answer to the question whether the graph attention mechanism can mitigate over-smoothing through a rigorous mathematical analysis.
>
> **Q2. Only homophily graphs are considered in the paper. In addition, heterophily datasets also should be considered to verify the overall performance of GNN as pointed out by recent publications. In addition, experiments on larger datasets will strengthen the significance of the contributions.**
>
> Thank you for your constructive suggestion. In line with the reviewer’s suggestion, we validate our theoretical findings on three heterophily datasets (Cornell, Texas and Wisconsin [1]) and a large-scaler dataset (Flickr [2]). The results are provided in the rebuttal pdf file.
>
> Our theoretical results are developed for generic graphs and thus hold for datasets exhibiting either homophily or heterophily and even those that are not necessarily either of the two.
>
> **Q3: What criteria were used to select the negative slope values in the experiment?**
>
> Thank you for the question. For LeakyReLU, the negative slope value can be selected within the range of 0 to 1, and the default value is 0.01. To illustrate its effects across various scenarios, we consider three distinct representative values: 0.01 for a small slope, 0.4 for a medium slope, and 0.8 for a large slope.
>
> **Q4: How sensitive are the hyperparameters (e.g. negative slope values) to the performance?**
>
> Thank you for the question. The sensitivity can be directly observed from Figure 1 in the paper, where for all negative slope values, oversmoothing persists and happens exponentially, although the degree of oversmoothing varies for different choices of nonlinearities.
>
> We appreciate your questions and comments very much. Please let us know for any further questions.
>
> ---
> **References**
>
> [1] Pei et al.. Geom-GCN: Geometric Graph Convolutional Networks. In ICLR 2020.
>
> [2] Zeng et al.. GraphSAINT: Graph Sampling Based Inductive Learning Method. In ICLR 2020.
>
> [3] Kenta Oono and Taiji Suzuki. Graph Neural Networks Exponentially Lose Expressive Power for Node Classification. In ICLR 2020.
>
> [4] Chen Cai and Yusu Wang. A note on over-smoothing for graph neural networks. In ICML Graph Representation Learning and Beyond (GRL+) Workshop, 2020.
>
> [5] Yimeng Min, Frederik Wenkel, and Guy Wolf. Scattering GCN: Overcoming oversmoothness in graph convolutional networks. In NeurIPS 2020.
>
> [6] Rusch et al.. Graph-coupled oscillator networks. In ICML 2022.

---

> > ### Comment · Reviewer_Ac2q · 2023-08-13
> >
> > The authors have effectively dealt with the issues I raised in my previous remarks. I'm pleased to elevate my rating and evaluation to WA.

---

### Official Review · Reviewer_QRK7 · 2023-07-06

**Soundness:** 3 good
**Presentation:** 4 excellent
**Contribution:** 3 good
**Rating:** 8
**Confidence:** 4

**Summary:**

The paper addresses the question of whether graph attention mechanism could prevent oversmoothing in Graph Neural Networks (GNN) and provides a negative result. Existing results on graph attention are available for 1-layer GNN. The current work analyzes the general case of multi-layered graph attention in the context of nonlinear, time-varying dynamical systems. Through a nicely presented rigorous theoretical analysis, the paper shows that (1) oversmoothing happens even for multi-layered attention based GNN (hence attention based mechanisms fundamentally does not change the aggregator operator) (2) oversmoothing happens at exponential rate and the analysis can be also generalized to GCNs. The numerical experiments demonstrate the validation of the claims made in the paper. The presented analysis could be useful in understanding other aspects of graph attention.

**Strengths:**

1. The paper addresses an important question of oversmoothing in multi-layered graph attention neural networks. It is very well written and the ideas across different sections are nicely connected. There is ample amount of intuition behind choice of various techniques used in the paper and the theorems/lemmas are well explained.

2. The ideas/techniques introduced in the paper are well developed, original and should be useful in understanding other aspects of attention based GNNs. I like that the authors provide remarks (lines 257) which address how the paper differs from previous techniques used in theoretical analysis of graph attention.

3. The results of the paper are of significance in understanding of GNNs in terms of developing an understanding of trajectory of matrices in graph attention and hence indirectly supporting methods like edge-dropping etc.

**Weaknesses:**

1. Some of the prior work on reducing oversmoothing [1] and impact of the current work for understanding them could be included as limitation for the current work. It would be great if authors could talk about if some of the techniques developed here applies to them.

[1]  Chen, Deli, et al. "Measuring and relieving the over-smoothing problem for graph neural networks from the topological view." Proceedings of the AAAI conference on artificial intelligence. Vol. 34. No. 04. 2020.

**Questions:**

Q1. Typo: The sub-indices for $W^{(k)}$ in recursive formulation of *Eq (6)* is flipped and should be - $j_k, j_{k+1}$ instead of $j_{k+1}, j_k$. This would match those in *Eq 5* case $W_{ji}^{(t)}$ for $j_{k+1}=i$. Same goes for the supplement. Please check that the proof that The proof the follows is alright.

Q2. In lines $286.287$, how is the existence of $q<1$ justified? I understand that this is an assumption in the *Theorem 1*, but could the authors shed light on nature of values $q$ takes since it dictates the convergence.

---

> ### Author Rebuttal · Authors · 2023-08-10
>
> We appreciate your thoughtful comments and positive assessment of our work. After carefully reviewing your feedback, below we provide answers to the comments you raised.
>
> **Q1: Some of the prior work on reducing oversmoothing [1] and impact of the current work for understanding them could be included as limitations for the current work. It would be great if authors could talk about if some of the techniques developed here apply to them.**
>
> This is an interesting point and is closely related to the discussion right after Theorem 1 (line 297-303). Based on our proof, a key property leading to oversmoothing is that the graph structure does not change across different layers. Our theory hence indirectly endorses empirical strategies such as edge-dropping or graph-rewiring, which modify the graph's connectivity structure to alleviate oversmoothing. In particular, AdaEdge proposed in [1] is a graph rewiring method.
>
>
> **Q2: Typo: The sub-indices for $W^{(k)}$ in recursive formulation of Eq (6) is flipped and should be $j_{k},j_{k+1}$ instead of $j_{k+1},j_{k}$. This would match those in Eq 5 case $W_{ji}^{(t)}$ for $j_{k+1}=i$. Same goes for the supplement. Please check that the proof follows is alright.**
>
> Thanks for the good catch! Indeed, the indices are flipped and it should be $W_{j_kj_{k+1}}$ instead. We verified that the rest of the proof had no issues.
>
> **Q3: In lines 286-287, how is the existence of $q<1$ justified? I understand that this is an assumption in Theorem 1, but could the authors shed light on the nature of values $q$ takes since it dictates the convergence.**
>
> Thank you for the question.
> - We note that $q$ is an arbitrary constant between $JSR(\tilde{\mathcal{M}}\_{\mathcal{G},\epsilon})$ and 1, which is guaranteed to exist by Lemma 6. The convergence rate dictated by $q$ (as mentioned by the reviewer) is thus determined by the joint spectral radius of the matrix set $\tilde{\mathcal{M}}\_{\mathcal{G},\epsilon}$.
> - However, as we explain in the paper (lines 305-306), computing or approximating the joint spectral radius for a given set of matrices is known to be hard in general [2]. It is thus insightful to derive bounds on the joint spectral radius. This is done in Proposition 2 in the paper, where we show that  $JSR(\tilde{\mathcal{M}}\_{\mathcal{G},\epsilon})$ is lower-bounded by the second largest eigenvalue of the normalized adjacency matrix ($D_{\deg}^{-1/2}AD_{\deg}^{-1/2}$).
>
> We appreciate your questions and comments very much. Please let us know for any further questions.
>
> ---
> **References**
>
> [1] Chen et al.. Measuring and relieving the over-smoothing problem for graph neural networks from the topological view. In AAAI 2020.
>
> [2] John Tsitsiklis and Vincent Blondel. The Lyapunov exponent and joint spectral radius of pairs of matrices are hard – when not impossible – to compute and to approximate. Mathematics of Control, Signals and Systems, 1997.

---

> > ### Comment · Reviewer_QRK7 · 2023-08-18
> > **Thank you for the rebuttal**
> >
> > The response from the authors addresses my questions.

---

### Author Rebuttal · Authors · 2023-08-10


We would like to thank the reviewers for carefully reading our paper and giving insightful comments. We have provided detailed responses to each of the reviews separately.

In the rebuttal pdf file, we also provide additional numerical results under the same experimental setup as in our paper. Specifically, we validate our theoretical findings on three heterophily datasets (Cornell, Texas and Wisconsin [1]) and a large-scale dataset (Flickr [2]), as suggested by one of the reviewers.

The authors


---
**References**

[1] Pei et al.. Geom-GCN: Geometric Graph Convolutional Networks. In ICLR 2020.

[2] Zeng et al.. GraphSAINT: Graph Sampling Based Inductive Learning Method. In ICLR 2020.

---

### Decision · Program_Chairs · 2023-09-21

**Decision:**

Accept (spotlight)

**Comment:**

The paper shows that contrary to popular belief, the graph attention mechanism cannot prevent oversmoothing and loses expressive power exponentially. Reviewers have noted the paper's great writing and the well-elaborated and innovative concepts and techniques it presents. The findings in the paper have substantial implications for advancing the understanding of GNNs and offer instructive and valuable insights into research on oversmoothing.